# BAYESIAN BI-CLUSTERING OF NEURAL SPIKING ACTIVITY WITH LATENT STRUCTURES

**Ganchao Wei**
Department of Statistical Science
Duke University
Durham, NC 27708, USA
`ganchao.wei@duke.edu`

## ABSTRACT

Modern neural recording techniques allow neuroscientists to obtain spiking activity of multiple neurons from different brain regions over long time periods, which requires new statistical methods to be developed for understanding structure of the large-scale data. In this paper, we develop a bi-clustering method to cluster the neural spiking activity spatially and temporally, according to their low-dimensional latent structures. The spatial (neuron) clusters are defined by the latent trajectories within each neural population, while the temporal (state) clusters are defined by (populationally) synchronous local linear dynamics shared with different periods. To flexibly extract the bi-clustering structure, we build the model non-parametrically, and develop an efficient Markov chain Monte Carlo (MCMC) algorithm to sample the posterior distributions of model parameters. Validating our proposed MCMC algorithm through simulations, we find the method can recover unknown parameters and true bi-clustering structures successfully. We then apply the proposed bi-clustering method to multi-regional neural recordings under different experiment settings, where we find that simultaneously considering latent trajectories and spatial-temporal clustering structures can provide us with a more accurate and interpretable result. Overall, the proposed method provides scientific insights for large-scale (counting) time series with elongated recording periods, and it can potentially have application beyond neuroscience.

## 1 INTRODUCTION

In neuroscience, identifying types of neurons is a longstanding challenge (Nelson et al., 2006; Bota & Swanson, 2007; Zeng, 2022). Some criteria based on features such as anatomical regions, genomics and synaptic connectivity have been proposed, and there are some Bayesian approaches to integrate these features (Jonas & Kording, 2015). On the other hand, the response pattern and interactions between neural populations may change over time, especially when the experiment stimuli changes (Pooresmaeili et al., 2014; Oemisch et al., 2015; Ruff & Cohen, 2016; Steinmetz et al., 2019; Cowley et al., 2020). However, these complex dynamical observations can often be broken down into simpler units, and it can be appropriate to assume static linear dynamics within chunks of the data. Moreover, it is usually appropriate and helpful to assume similar linear dynamics can be shared by different epochs (Zoltowski et al., 2020; Glaser et al., 2020b). Here, we consider the problem of how to identify both spatial and temporal clusters of neural spikes.

The modern techniques such as the high-density probes (Jun et al., 2017; Steinmetz et al., 2021; Marshall et al., 2022) allow us to obtain large-scale multi-electrode recordings from multiple neurons across different anatomical regions over an elongated session. Several models have been developed to extract shared latent structures from simultaneous neural recordings, assuming that the activity of all recorded neurons can be described through common low-dimensional latent states (Cunningham & Yu, 2014; Gao et al., 2017). These approaches have proven useful in summarizing and interpreting high-dimensional population activity. Inferred low-dimensional latent states can provide insight into the representation of task variables (Churchland et al., 2012; Mante et al., 2013; Cunningham & Yu, 2014; Saxena & Cunningham, 2019) and dynamics of the population itself (Vyas et al., 2020). Many existing approaches are based on linear dynamical system (LDS) model (Macke et al., 2011),

which is built on the state-space model and assumes latent factors evolve with static linear dynamics. Although assuming static linear dynamics over time can be valid in some tasks and in small chunks of experiment, the assumption is not generally appropriate. To tackle the nonlinear dynamics, some variants of the LDS, such as switching-LDS (SLDS, Ghahramani & Hinton (2000); Fox (2009); Fox et al. (2008a); Murphy (2012)) and recurrent-SLDS (RSLDS, Linderman et al. (2017; 2019)) have been proposed. The non-parametric Gaussian process factor analysis (GPFA) model (Yu et al., 2009) and its variants provide a more flexible way to model non-linaer neural data, although most these methods assume independent GP and doesn't allow for interactions between latent factors. Recently, (Cai et al., 2023) proposed the dependent GP method using the kernel convolution framework (KCF, Boyle & Frean (2004); Álvarez & Lawrence (2011); Sofro et al. (2017)), but their method may not scalable for elongated neural recordings. Several methods have been developed and implemented to analyze multiple neural populations and their interactions (Semedo et al., 2019; Glaser et al., 2020b), as the interactions may occur in low-dimensional subspace (Stavisky et al., 2017; Kaufman et al., 2014; Semedo et al., 2019). But the neural populations are prespecified, and the spatio-temporal clustering structures of neural data haven't been evaluated systematically in general.

The clustering of neural spikes is an important and long-lasting problem, and people put a lot effort into developing methods to uncover patterns from the complex data. The neural spiking activity is essentially a point process, and there are some methods for finding clusters in point process by, such as, Dirichlet mixture of Hawkes process (Xu & Zha, 2017), mixture of multi-level point process and (Yin et al., 2021) and group network Hawkes process (Guanhua Fang & Guan, 2023). All these methods are general and have wide rage of applications, but the defined "clusters" may not directly related to underlying mechanisms and may lose some scientific insights. To deal with this problem, some methods like mixPLDS (Buesing et al., 2014) and recent mixDPFA method (Wei et al., 2022; 2023) try to cluster neurons according to their latent structures, by using the mixture of LDS model. These approach provides a more interpretable and accurate way to clusters neurons, and may be useful for identifying "functional populations" of neurons. However, these methods assume the static linear dynamics and don't allow for the interactions between neural populations, which can limit the usage of these methods, and may bias or even fail the detection of neural populations when considering the elongated recordings, especially under different experiment conditions. On the other hand, for the clustering structures in terms of temporal states, most methods are developed based on the SLDS, by modeling the nonlinear dynamics with local linear pattern. Instead of clustering based on linear dynamics, D'Angelo et al. (2023) recently tried to cluster the experiment periods based on the distributions of spiking amplitude, using a nested formulation of mixture of finite mixtures model (MFMM), i.e., exploiting the generalized MFMM (gFMFMM, Frühwirth-Schnatter et al. (2021)) prior with common atom model (Denti et al., 2023).

In this research, we develop a bi-clustering method to cluster neural spikes both spatially (to give subject clusters) and temporally (to give state clusters), according to the latent structures of these neurons (Figure 1A). The neural population is defined via private low-dimensional latent trajectories as in mixPLDS (Buesing et al., 2014) or mixDPFA (Wei et al., 2022; 2023). For the state clusters, we assume the linear dynamics can be shared across different chunks and the state clustering structures are defined by local linear manner as in (R)SLDS. Neurons in each population is assume to have private latent trajectories, but all time series are assumed to switch between different states synchronously, to use the information from all observations. Besides extending the previous clustering method like mixDPFA to simultaneously detect the state cluster, the proposed bi-clustering method also allow for interactions between neural populations and non-stationary dynamics for neural response, using similar idea from (Glaser et al., 2020a). Simultaneously considering all these effects in the proposed bi-clustering method is necessary, since incorrect population assignments can lead to biased and inconsistent inference on the latent structure (Ventura, 2009). On the other hand, these flexibility allows for more accurate estimate of latent trajectories, and hence will lead to a more accurate estimates of the subject clustering structure.

To flexibly infer the bi-clustering structure, we model them non-parametrically to avoid prespecifying the number for subject and state clusters. Specifically, the subject clustering structure is modeled by a mixture of finite mixtures model (MFMM, Miller & Harrison (2018)) of latent trajectories and the state clustering structure is modeled by a sticky Hierarchical Dirichlet Process Hidden Markov Model (sticky- HDP-HMM, Fox et al. (2008b)). The posteriors of model parameters are sampled using a Markov Chain Monte Carlo (MCMC) algorithm, where the Pólya-Gamma data augmentation technique (Polson et al., 2013) is used to handle the counting observations for neural spikes.

The rest of this paper is structured as follows. In Section 2, we introduce the bi-clustering method for time series with counting observations, and provide brief explanations of the MCMC algorithm to sample posterior distributions of parameters. After validating the proposed bi-clustering method with a synthetic dataset in Section 3, we then apply our method to analyze multi-regional experimental recordings from a behaving mouse under different experiment settings in Section 4. Finally, in Section 5, we conclude with some final remarks and highlight some potential extensions of our current model for future research.

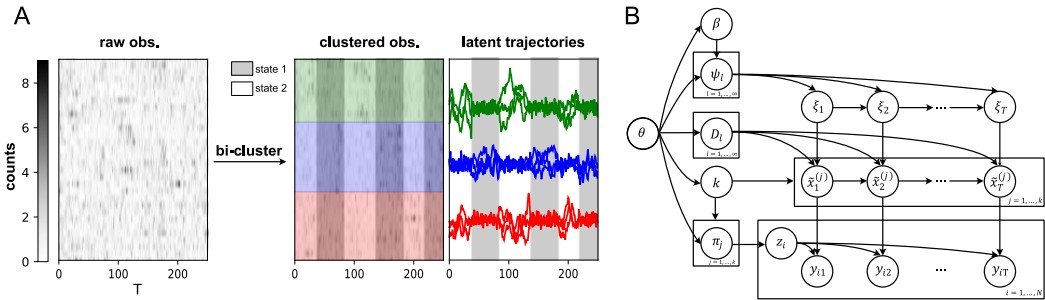

Figure 1: **Model overview. A**. The goal for our proposed model is to do clustering both spatially and temporally (i.e. "bi-clustering") for neural spike data (time series data with counting observations), according to their latent structures. The neural spiking counts are determined by a low dimensional latent factors, specific to the spatially subject clustering assignment (e.g. green, blue and red). On the other hand, all neurons are assumed to switch between states synchronously, and are temporally clustered according to different states of linear dynamics (e.g. gray and white). **B**. Graphical model of the proposed bi-clustering model. All prior parameters are summarized as $\theta$, and parameters such as $d_i$ and $c_i$ dropped for simplicity.

## 2 BI-CLUSTERING METHOD FOR NEURAL SPIKES

In this section, we introduce a bi-clustering model for neural spiking activity, i.e., the time series data with counting observations. The goal for the proposed model is to cluster neural spikes both spatially (for subject cluster) and temporally (for state cluster), based on the multi-population and -state latent structures. To flexibly capture the clustering structures, we build model non-parametrically. The graphical representation of the model is summarized in Figure 1B. After introducing the model, we briefly describe how a MCMC algorithm is used to infer model parameters.

### 2.1 MULTI-POPULATION AND -STATE LINEAR DYNAMIC MODEL

Assume we can observe spiking activity of $N$ neurons up to recording length $T$. Denote the number of counts for neuron $i \in \{1, \ldots, N\}$ at time bin $t \in \{1, \ldots, T\}$ as $y_{it} \in \mathbb{Z}_{\geq 0}$, and the cluster indicator of neuron $i$ as $z_i$ (i.e. the "subject indicator"). Assume the count $y_{it}$ follows a negative-binomial distribution, where the log-mean response is modeled by a linear combination of subject baseline $d_i$, population baseline $\mu_t^{(z_i)}$ and $p-$dimensional latent factor $\boldsymbol{x}_t^{(z_i)} \in \mathbb{R}^p$ (Here we assume all populations have the same latent dimension for convenience). In other words, the observation equation is as follows:

$$y_{it} \sim \text{NB}(r_i, \mu_{it})$$
$$\log \mu_{it} = d_i + \mu_t^{(z_i)} + \boldsymbol{c}_i' \boldsymbol{x}_t^{(z_i)} \tag{1}$$

, where $\text{NB}(r, \mu)$ denotes the negative-binomial distribution (NB) with mean $\mu$ and variance $\mu + \mu^2/r$, and $\boldsymbol{c}_i \sim \mathcal{N}(\boldsymbol{0}, \boldsymbol{I}_p)$. The NB distribution can be replaced by a Poisson distribution when it's appropriate to assume equi-dispersion, for ease of model inference. If we further denote $\tilde{\boldsymbol{x}}_t^{(j)} = (\mu_t^{(j)}, \boldsymbol{x}_t'^{(j)})'$ and $\tilde{\boldsymbol{c}}_i = (1, \boldsymbol{c}_i')'$, then $\log \mu_{it} = d_i + \tilde{\boldsymbol{c}}_i' \tilde{\boldsymbol{x}}_t^{(z_i)}$. To save words and notations, if not specified, we refer $\tilde{\boldsymbol{x}}_t^{(j)}$ as "latent factor" for cluster $j$, which also includes the population baseline.

Although each neural population is modeled with private latent factors, there usually exist some interactions between clusters (Musall et al., 2019; Stringer et al., 2019), and these interactions can change over time (Ruff & Cohen, 2016; Steinmetz et al., 2019; Cowley et al., 2020), especially when the external condition changes. On the other hand, interactions between neural populations and receiving common inputs for all neurons suggest that neurons in different clusters may synchronize the response states over time. Therefore, to allow for the interactions between populations and model the synchronous state switching, we stack the latent factors for all clusters together, and assume all latent factors evolve in a conditional linear manner, given the discrete latent states $\xi_t$ shared across the cluster, as in Glaser et al. (2020b). In other words, the state clustering structure is defined by the local linear dynamics of latent factors, by assuming complex dynamics can be decomposed into simple linear unit and the small chunks of the neural response can be sufficiently described by the LDS model. Specifically, assume there are $k$ unique clusters, i.e., $|\{z_i\}_{i=1}^N| = k$, the cluster-stacked latent factors (including population baseline) is denoted as $\tilde{\boldsymbol{X}}_t = (\tilde{\boldsymbol{x}}_t'^{(1)}, \ldots, \tilde{\boldsymbol{x}}_t'^{(k)})' \in \mathbb{R}^{k(p+1)}$. To capture temporal dynamics of the data, we further put AR(1) structure onto the latent factors $\tilde{\boldsymbol{X}}_t$. In other words, given the discrete latent state at $t$ as $\xi_t$ (i.e. the "state indicator" shared across the subject cluster), $\tilde{\boldsymbol{X}}_t$ is assumed evolve linearly with a Gaussian noise as follows:

$$\tilde{\boldsymbol{X}}_{t+1} = \boldsymbol{b}_{\xi_t} + \boldsymbol{A}_{\xi_t}\tilde{\boldsymbol{X}}_t + \boldsymbol{\epsilon}_{\xi_t} \tag{2}$$

, where $\boldsymbol{\epsilon}_{\xi_t} \sim \mathcal{N}(\boldsymbol{0}, \boldsymbol{Q}_{\xi_t})$ and $\tilde{\boldsymbol{X}}_1 \sim \mathcal{N}(\boldsymbol{0}, \boldsymbol{I}_{k(p+1)})$. Here, the dynamic parameters $(\boldsymbol{b}_{\xi_t}, \boldsymbol{A}_{\xi_t}, \boldsymbol{Q}_{\xi_t})$ summarize dynamics (state changes) within and across population. To make the model identifiable, we further assume 1) $\tilde{\boldsymbol{x}}^{(j)}$ is zero-centered, i.e. $\sum_{t=1}^T \tilde{\boldsymbol{x}}_t^{(j)} = \boldsymbol{0}$ and 2) $\tilde{\boldsymbol{x}}_{1:T}^{(j)}\tilde{\boldsymbol{x}}_{1:T}'^{(j)}$ is diagonal, where $\tilde{\boldsymbol{x}}_{1:T}^{(j)} = (\tilde{\boldsymbol{x}}_1^{(j)}, \ldots, \tilde{\boldsymbol{x}}_T^{(j)}) \in \mathbb{R}^{(p+1) \times T}$ (Fokoué & Titterington, 2003). The identifiability issue under signed-permutation is handled by alignment to samples in early stage of the chain. For more details on model constraints, see Section A.1. In summary, given the neuron $i$ belonging to cluster $z_i = j$, the counting series is generated by a negative-binomial linear model $\mathcal{M}$ defined in equation 1 as $(y_{i1}, \ldots, y_{iT})' \sim \mathcal{M}(d_i, \boldsymbol{c}_i, \tilde{\boldsymbol{x}}_{1:T}^{(j)})$, where the prior for $\tilde{\boldsymbol{x}}_{1:T}^{(j)}$ is denoted as $\mathcal{H}$. The within- and between-population linear dynamics at $t-$th step are captured by dynamical parameters $(\boldsymbol{b}_{\xi_t}, \boldsymbol{A}_{\xi_t}, \boldsymbol{Q}_{\xi_t})$, where $\xi_t$ is the state indicator at $t$. To do clustering both spatially (subject cluster) and temporally (state cluster) in a flexible way, we model each of these two clustering structures non-parametrically as follows.

## 2.2 SUBJECT CLUSTERING MODEL

Since the number of neural populations is finite but unknown, we put prior on number of subject cluster $|\{z_i\}_{i=1}^N| = k$ as in (Wei et al., 2023), which leads to the mixture of the finite mixtures model (MFMM) as follows:

$$
\begin{aligned}
K &\sim f_k, & f_k \text{ is a p.m.f. on} \{1, 2, \ldots\}, \\
\boldsymbol{\pi} = (\pi_1, \ldots, \pi_k) &\sim \text{Dir}_k(\gamma, \ldots, \gamma) & \text{given } K = k, \\
z_1, \ldots, z_N &\overset{i.i.d.}{\sim} \boldsymbol{\pi} & \text{given } \boldsymbol{\pi}, \\
\tilde{\boldsymbol{x}}_{1:T}^{(1)}, \ldots, \tilde{\boldsymbol{x}}_{1:T}^{(k)} &\overset{i.i.d.}{\sim} \mathcal{H} & \text{given } k, \\
(y_{i1}, \ldots, y_{iT})' &\sim \mathcal{M}(d_i, \boldsymbol{c}_i, \tilde{\boldsymbol{x}}_{1:T}^{(z_i)}) & \text{given } d_i, \boldsymbol{c}_i, \tilde{\boldsymbol{x}}_{1:T}^{(z_i)}, z_i, \text{for } i = 1, \ldots, N,
\end{aligned}
\tag{3}
$$

, where p.m.f denotes the probability mass function. By using the MFMM, we can integrate the field knowledge about the number of clusters into our analysis, by specifying the $f_k$. In the analysis of this paper, we assume $k$ follows a geometric distribution, i.e., $k \sim \text{Geometric}(\zeta)$ with p.m.f. defined as $f_k(k \mid \zeta) = (1-\zeta)^{k-1}\zeta$ for $k = 1, 2, \ldots$, and $\gamma = 1$. For general use of the proposed method to some problems where the number of subject cluster number can potentially grow to infinity, using the mixture model such as the Dirichlet process mixtures model (DPMM) maybe conceptually more appropriate. See Miller & Harrison (2018) for more detailed discussion.

## 2.3 STATE CLUSTERING MODEL

For state clustering structure, as the number of states can potentially shoot to infinity, we model the discrete state $\xi_t$ by a sticky Hierarchical Dirichlet Process Hidden Markov Model (sticky-HDP-

HMM) proposed by (Fox et al., 2008b) to encourage the re-occurrence of states as follows:

$$
\begin{aligned}
\beta &\sim \text{GEM}(\eta), \\
\psi_l &\overset{i.i.d.}{\sim} \text{DP}(\alpha + m, \frac{\alpha\beta + m\delta_l}{\alpha + m}) \\
\xi_t &\sim \psi_{\xi_{t-1}}, \\
(\boldsymbol{b}_l, \boldsymbol{A}_l, \boldsymbol{Q}_l) &\overset{i.i.d.}{\sim} \mathcal{S}, && \text{for } l = 1, 2, \ldots, \\
\tilde{\boldsymbol{X}}_{t+1} &\sim \mathcal{N}(\boldsymbol{b}_{\xi_t} + \boldsymbol{A}_{\xi_t}\tilde{\boldsymbol{X}}_t, \boldsymbol{Q}_{\xi_t}) && \text{for } t = 1, \ldots, T
\end{aligned}
\tag{4}
$$

, where GEM denotes the stick breaking process (Sethuraman, 1994), $\delta_i$ denotes the indicator function at index $i$, DP denotes the Dirichlet process and $\mathcal{S}$ denotes the normal-inverse-Wishart prior of $(\boldsymbol{b}_l, \boldsymbol{A}_l, \boldsymbol{Q}_l)$. The details of $\mathcal{S}$ can be found in the appendix A.1, when introducing the MCMC algorithm. The sticky HDP-HMM extends the HDP-HMM with a "sticky" parameter $m > 0$ to encourage longer state duration, and hence can handle the rapid-switching problem to some degree. Some more careful methods for modeling the state duration and state transition is further discussed in the Section 5.

## 2.4 MODEL INFERENCE

We do Bayesian inference on the proposed bi-clustering model by an efficient MCMC algorithm. In each sampling iteration, there are approximately four steps: 1) sample dynamical latent factors $\tilde{\boldsymbol{x}}_{1:T}^{(z_i)}$, 2) sample remaining subject-specific parameters in observation equation equation 1, including subject baseline $d_i$, factor loading $\tilde{\boldsymbol{c}}_i$ and dispersion $r_i$, 3) sample the temporal states $\xi_t$ and corresponding dynamical parameters $(\boldsymbol{b}_l, \boldsymbol{A}_l, \boldsymbol{Q}_l)$ for each sampled state, and 4) sample the subject cluster indices $z_i$. The details of sampling procedures can be found in the appendix Section A.1, and we briefly introduce the key sampling method for each step here.

In step 1), the full conditional distribution of latent factors $\tilde{\boldsymbol{x}}_{1:T}^{(j)}$ is equivalent to the posterior distribution of the negative-binomial dynamic GLM (NB-DGLM), which has no closed form. However, the NB distribution falls within a Pólya-Gamma (PG) augmentation scheme (Polson et al., 2013; Windle et al., 2013; Linderman et al., 2016), therefore we can sample them in closed form by introducing the PG augmented variables. Conditioning on the auxiliary variables $\omega_{it}$, the transformed "effective" observations $\hat{y}_{it}$ has Gaussian likelihood, and hence we can sample the posterior of $\tilde{\boldsymbol{x}}_{1:T}^{(j)}$ using the forward-filtering-backward-sampling (FFBS, Carter & Kohn (1994); Frühwirth-Schnatter (1994)) algorithm. For Poisson observation model, we can treat the data as coming from the NB distribution, use the samples as proposal and add one more Metropolis-Hasting (MH) step to accept or reject the proposal. In Poisson case, the dispersion $r_i$ becomes the tuning parameter to achieve desirable acceptance rate (Wei et al., 2022; 2023).

In step 2), the sampling of $d_i$ and $\tilde{\boldsymbol{c}}_i$ is regular NB regression problem, and we again use the PG data augmentation technique to sample them. The dispersion parameter $r_i$ is updated via a Gibbs sampler, using the method described in Zhou et al. (2012), as the gamma distribution is the conjugate prior to the $r_i$ under the compound Poisson representation.

In step 3), the discrete states $\xi_t$ are sampled by a weak-limit Gibbs sampler for sticky HDP-HMM as in Fox et al. (2008b). The weak-limit sampler constructs a finite approximation to the HDP transitions prior with finite Dirichlet distributions, as the infinite limit converges in distribution to a true HDP. Given the latent factors $\tilde{\boldsymbol{x}}_{1:T}^{(j)}$ and state indicator $\xi_t$, we can update dynamical parameters $(\boldsymbol{b}_l, \boldsymbol{A}_l, \boldsymbol{Q}_l)$ for each state separately in closed form.

In step 4), given the parameters in observation equation equation 1, we sample the subject cluster indices $z_i$ using the algorithm for MFMM proposed by Miller & Harrison (2018), which is analogous to the partition-based algorithm for DPMM (Neal, 2000). The label switching issue of $z_i$ is handled by the Equivalence Classes Representatives (ECR) algorithm (Papastamoulis & Iliopoulos, 2010), by using the sample from early stage of the chain as pivot allocation. When sampling the subject clustering assignments $z_i$ in such a high dimensional time series data with large $T$, evaluating the full likelihood given samples of $\boldsymbol{c}_i$ as in Gaussian MFA (Fokoué & Titterington, 2003) leads to a poor mixing chain. Instead, we evaluate the marginalized likelihood by integrating out the subject-

specific loading $c_i$, similar to Wei et al. (2023). The marginalized likelihood is evaluated by Laplace approximation.

The Python implementation of the NB and Poisson bi-clustering model is available in `https://github.com/weigcdsb/bi_clustering` and supplementary material. The additional details for MCMC sampling can be found in appendix Section A.1.

## 3 SIMULATION

To validate and illustrate the proposed bi-clustering method, we simulate neural spikes from the NB bi-clustering generative model defined in observation equation 1 and system equation 2. In this simulation, we generate 3 clusters with 10 neurons in each cluster ($N = 30$ in total). The recording length is $T = 500$ and the dimension for $x_{1:T}^{(j)}$ are all $p = 2$. For each neuron, the individual baseline is generated by $d_i \sim N(0, 0.5^2)$, the factor loading is generated by $c_i \sim N(\mathbf{0}, \mathbf{I}_2)$ and dispersion are all $r_i = 10$. For latent factors of these three clusters $\{\tilde{\boldsymbol{x}}_{1:T}^{(j)}\}_{j=1}^3$, they are generated from two discrete states, and the state indicator $\xi_t = 1, 2$ is generated from a semi-Markov chain (Sansom & Thomson, 2001; Yu, 2010), to encourage longer state duration. These states correspond two sets of linear dynamics: 1) independent state, where $\boldsymbol{A} \in \mathbb{R}^9$ is diagonal and 2) interactive state, where $\boldsymbol{A}$ is a random rotation of an orthogonal matrix, and hence there are interactions between clusters. The bias term is $\boldsymbol{b} = \mathbf{0}$ and noise covariance is $\boldsymbol{Q} = \boldsymbol{I}_9 \cdot 10^{-2}$ for both states.

We then apply the proposed bi-clustering methods to the simulated data, by setting the maximum number of states be 10 for the weak-limit sampler. Here, we run a MCMC chain for 10,000 iterations, starting from 1 subject cluster and 10 uniformly distributed temporal states. The results shown here summarize the posterior samples from iteration 2500 to 10,000. The trace plots for several parameters are shown in appendix (Figure 4), and they don't suggest significant convergence issues. First, to evaluate the inferred clustering structure, we check the similarity matrices for both state (Figure 2A) and subject cluster (Figure 2B). The entry $(i, j)$ for a similarity matrix is the posterior probability that data points $i$ and $j$ belong to the same cluster. Both subject and state similarity matrices are sorted according to true clustering assignments, and hence if the algorithm can recover the simulated cluster structures, the diagonal blocks will have high posterior probability, which is the pattern shown in Figure 2A and 2B. The histograms of posterior samples (Figure 2C)show that our method can successfully recover the number of subject and state cluster. To represent the clustering structure in temporal state more intuitively, we also provide the single point estimates of $\xi_t$ (Figure 2D) by maximizing the posterior expected adjusted Rand index (maxEPAR, Fritsch & Ickstadt (2009)), which performs better than other points estimates such as the MAP estimates. All these results show that we can successfully recover the clustering structures on spatial and temporal dimension, including the number of clusters for each.

One the other hand, the advantage for proposed bi-clustering method is that it can simultaneously provide unbiased estimates of the latent trajectories for each cluster, which can be very helpful for scientific interpretation and insights. Here, we show the latent trajectories for the cluster that most neuron 1-10 belong to in Figure 2E (subject 1-10 also forms the a maxPEAR subject cluster). The samples for $\boldsymbol{x}_t^j$ (excluding $\mu_t^j$) in Figure 2E are rotated by a single matrix, to match the posterior mean to ground truth (the rotation matrix is found least square between posterior mean and ground truth). The trace plots for L2/ Frobenius norms of latent trajectories (show both raw and rotated traces for $\boldsymbol{x}_t^j$) for each detected cluster is shown in Figure 4C-D. The fitting results show that simultaneously considering subject and state clustering structure is necessary for estimation of latent structures. We also show the performance of Poisson bi-clustering model (i.e. replace the NB distribution in observation equation 1 by Poisson distribution). The clustering results are summarized by similarity matrices (Figure 2F) and maxPEAR estimate of state (the third bar in Figure 2D). Since in this simulation example, the over-dispersion is not severe ($r_i = 10$), the Poisson version can also recover the true clustering structure, but with some bias in the estimation of latent trajectories. However, for data with large over-dispersion (which is common for real data), the wrong assumption on equi-dispersion will hugely influence the clustering structures, and it would be necessary to use the more flexible NB bi-clustering model.

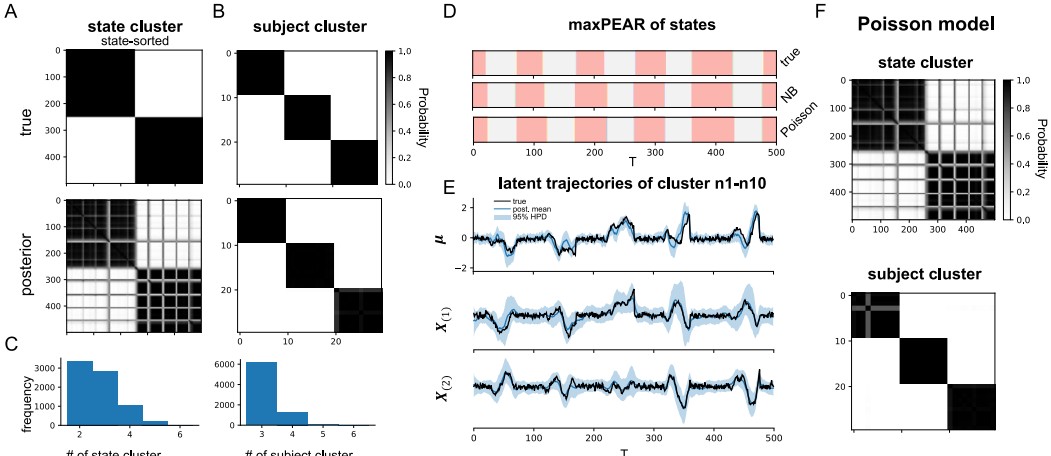

Figure 2: **Simulations.** Here, we show the results for posterior samples from iteration 2500 to 10,000 for each MCMC chain on the simulated dataset. These results are from NB bi-clustering model if not specified. **A**. The posterior similarity matrix for temporal states are ordered according to ground true states, representing the inferred clustering structures relative to ground truth. **B**. Spatially, the similarity matrix for subject are ordered according true subject clusters. **C**. The histograms of posterior samples on number of state cluster (true = 2) and subject cluster (true = 3). **D**. The max-PEAR estimates (point estimates) of the discrete states for NB and Poisson bi-clustering model, comparing to the true temporal states. **E**. The inferred latent trajectories for the detected cluster that most subject 1-10 belong to. Here $\boldsymbol{\mu}$ denotes $\mu_{1:T}$, $\boldsymbol{X}_{(i)}$ denotes $i$-th row of $\boldsymbol{x}_{1:T}$, and samples of $\boldsymbol{X}_{(i)}$ are transformed by a single matrix, to match posterior mean and ground truth (by least square). The black lines are truths, blue lines are posterior means and shaded light blue regions are 95% highest posterior density (HPD) regions. **F**. The similarity matrices of state and subject cluster for Poisson bi-clustering model, sorted using the same order as in panel **A** and **B** respectively.

## 4 APPLICATION

We then apply our bi-clustering method to Allen Institute Visual Coding Neuropixels dataset. The dataset contains neural spiking activity from multiple brain regions of an awake mouse, under different visual stimuli. See Siegle et al. (2021) for more detailed data description. Here, we use the electrophysiology session 719161530 to investigate the bi-clustering structures of neurons from three anatomical sites, under three consecutive experimental epochs. After excluding neurons having less than 1Hz response rate, 78 neurons are contained in the following analysis. Among these neurons, 37 originate from the hippocampal CA1, 20 from the lateral posterior nucleus of the thalamus (LP), and 21 from the primary visual cortex (VISp). The neural spikes are recorded when the mouse is exposed to three consecutive visual stimuli : spontaneous (S, durates 30.025s), natural movie (N, durates 300.251s) and again spontaneous (S, durates 30.025s). These three epochs are chosen, according to the hypothesis that spontaneous connectivity patterns in visual cortex are shaped by habitual coactivations produced by natural visual stimulation (Fiser et al., 2010; Harmelech & Malach, 2013; Sadaghiani & Kleinschmidt, 2013), and the similarity in connectivity patterns between spontaneous and natural movie are found in human visual cortex (Wilf et al., 2017). Here, we rebin the data with 500ms, and hence $T = 720$. For formal application, we may need a smaller bin size for the higher resolution. The binned spiking counts for these 78 neurons are shown in Figure 3A.

Then, we fit the data with both NB bi-clustering and Poisson bi-clustering model, and run two independent chains for each. The results from all four chains can be found in the Section A.3, Figure 6. Although the formal analysis requires us to tune some parameters such as latent dimension $p$ and the sticky parameter $m$ in equation 4, we here run 10,000 iterations using $p = 2$ and $m = 10$, simply to illustrate the usage of proposed method on real data. Since these neurons come from three brain regions, we set the prior for number of subject cluster as $k \sim \text{Geometric}(0.415)$ , such that

$P(k \leq 3) = 0.8$. The trace plots for several parameters are provided in appendix (Figure 5B-C), which don't suggest significant convergence issues.

For subject clustering structure, the NB bi-clustering model detects around 10 clusters (histogram and trace in Figure 5A-B), and the posterior similarity matrix sorted by maxPEAR estimate is shown in Figure 3C-i. Generalluy, the method detects a large neural population with a high "confidence", with several weak clusters. We further sort the similarity matrix according to the anatomical labels, to examine the relationship between subject clustering results and anatomy (Figure 3-ii). The re-sorted result show that most neurons of the detected largest cluster come from CA1, while some neurons in LP and VISp are also included. Moreover, although most identified subject clusters are neurons from the same anatomical area, there are some mismatches between these two criteria. Especially, some neurons in CA1 are grouped into the same cluster with neurons in VISp, and also between LP and VISp. This may imply that there are some "functional interactions" between CA1 and VISp, and LP and VISp. We also compare the subject clustering results from Poisson bi-clustering model and mixDPFA, and sort the similarity matrices using the same order as in Figure 3C-ii. When assuming the equi-dispersion and fit the Poisson bi-clustering model, there are more mismatches between the detected clusters and anatomy (Figure 3C-iii), which may suggest some spurious interactions are detected when ignoring the over-dispersion. The mixDPFA assumes Poisson distributed spikes and further ignores the potential state changes along the time. The detected clustering structures are more noisy, and especially can hardly distinguish between VISp and LP (Figure 3C-iv). The results are consistent with previous finding of the mixDPFA, which shows that the neural population may change under different experimental settings, if the static dynamics is assumed (Wei et al., 2023). Overall, these results suggest that it is necessary to consider the over-dispersion and time-varying nonlinear dynamics, to obtain unbiased estimate of clustering structures.

For the state clustering structures (histogram and trace in Figure 5A-B), the algorithm detects around 13 clusters, and we show the similarity matrix (Figure 3B) and the maxPEAR estimate (Figure 3D). The stage changes faster for spontaneous (S) than natural movie (N), but the temporal states don't show a clear pattern as in subject cluster in NB-biclustering model. This may suggest similarity between spontaneous and natural visual stimulation for temporal response, in addition to spatial connectivity pattern (Wilf et al., 2017). On the other hand, the Poisson bi-clustering model suggests the earlier and later neurons have different stages. But since the bin size is relatively large in current implementation, the over-dispersion may be significant and the results here may be spurious by ignoring the over-dispersion. Further application to data with "ground-truth" states, such as sleep/ wake, may help us study effects of over-dispersion in different models.

Finally, we also show the details of the largest maxPEAR subject cluster (traces of L2 norm for latent trajectoreis in Figure 5C). The largest maxPEAR cluster has 21 neurons, which contains 9 neurons from CA1, 7 neurons from LP and 5 neurons from VISp. The spiking counts of these neurons (Figure 3E) may suggest periodic pattern, i.e., alternating strong and weak response, in the middle portion of the natural movie epoch. The observed pattern is captured by the latent trajectories that most of these neurons belong to (Figure 3E).

## 5 DISCUSSION

In this paper, we introduce a Bayesian nonparametric method to cluster the neural spiking activity spatially and temporally, according to the latent structures (trajectories) for each neural population. Compared to other clustering method for time series (e.g. distance-based methods), the clustering structures defined by latent trajectories can provide scientific insights for the large-scale complicated time series data. Moreover, simultaneously consider the subject and state clustering structures can provide us unbiased and consistent estimates of latent structures (e.g. trajectories) and vice versa.

Although the proposed method can successfully bi-cluster the neural spikes, there are some potential improvements. First, the subject clustering structures are modeled by MFMM, which consider the nature for number of neural populations. However, the uncertainty of clustering results can be large in some cases, and hence it may be better to consider the generalized MFMM (gMFMM), which can provide greater efficiency in the cluster estimation (Frühwirth-Schnatter et al., 2021). Moreover, the common atom specification of gMFMM (Denti et al., 2023; D'Angelo et al., 2023) can provide flexiblity in partitions estimations, resolve the degeneracy isuues and more importantly can allow us borrow information from different neural populations. Second, we currently pre-specify and

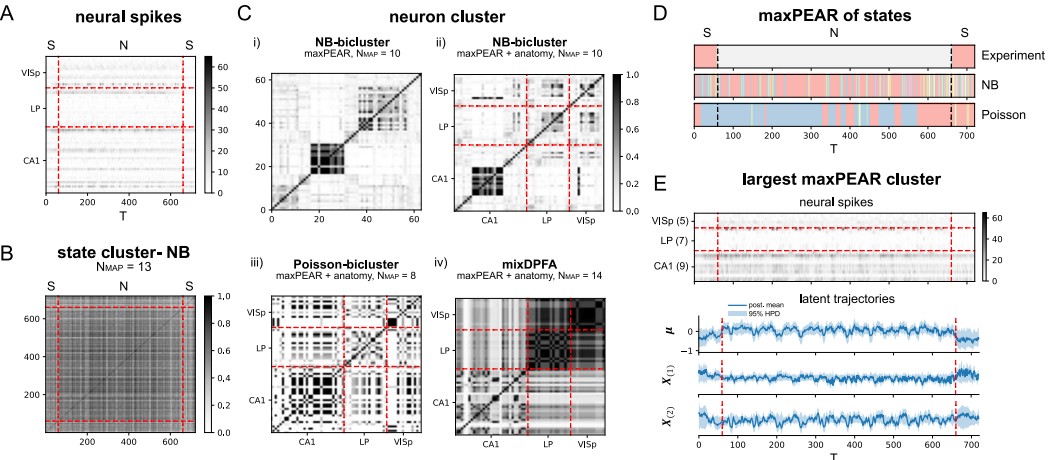

Figure 3: **Application in multi-regional neural data under different experiment epochs. A.** Here, we apply our method to multi-regional Neuropixels data, which contains neural spikes from 3 regions (CA1, LP and VISp) across 3 periods with different visual stimuli: spontaneous (S), natural movie (N) and spontaneous (S). The results from iteration 2500 to 10,000 for each chain are shown here. **B.** The similarity matrix of state cluster for NB bi-clustering model. **C.** The similarity matrices of neuron cluster sorted by maxPEAR estimate for NB bi-clustering model (NB-maxPEAR, upper-left). The clustering results sorted by both NB-maxPEAR and anatomical sites for three different clustering models (NB bi-clustering, Poisson bi-clustering and mixDPFA) are shown here for comparison. **D.** The maxPEAR estimates of the discrete states for NB and Poisson bi-clustering model. **E.** The largest maxPEAR cluster contains 9 neurons from CA1, 7 neurons from LP and 5 neurons from VISp. The upper panel shows the observed neural spikes. The lower panel shows the latent trajectories that most these neurons belong to, where the blue lines are posterior means and shaded light blue regions are 95% HPD regions.

assume all clusters share the same dimension of latent factors $p$ for convenience. However, this assumption may be inappropriate for real data application, and the method can be more flexible to infer $p$ at the same time. Previously, Wei et al. (2023) sample the latent dimension by a birth-and-death MCMC (BDMCMC) (Fokoué & Titterington, 2003; Stephens, 2000) with the marginalized likelihood, which requires very little mathmatical sophistication and is easy for interpretation. Some other methods, such as putting multiplicative Gamma process prior (Bhattacharya & Dunson, 2011), multiplicative exponential process prior (Wang et al., 2016) and Beta process priopr (Paisley & Carin, 2009; Chen et al., 2010) and Indian Buffet process prior (Knowles & Ghahramani, 2007; 2011; Ročková & George, 2016) may also be useful. Third, when clustering the temporal state, we tried to use the sticky-HDP-HMM to handle the rapid-switching issue. However, the method restrict to geometric state duration and doesn't allow for learning state-specific duration information. When applying to the Neuropixels data, the state looks change fast. This may suggest that we need to model the state duration more carefully, for example, by HDP-HSMM (Johnson & Willsky, 2013). Moreover, neither sticky-HDP-HMM nor HDP-HSMM allow the transition of discrete latent state $\xi_t$ to depend on latent trajectories $\tilde{x}_t$. Therefore, it may be possible to combine idea of recurrent HMM (Linderman et al., 2017) with HDP-HSMM, which may lead to some method like HDP-recurrent-HSMM, for instance. Finally, although the MCMC algorithm developed here is quite efficient, a deterministic approximation of MCMC, such as variational inference may be more computationally efficient and can be more attractive for scientific application.

To sum up, as the scale of neural spiking data becoming large both spatially and temporally, understanding the latent structures of multiple populations under different conditions can be a major statistical challenge. Here, we provide a way to extract spatio-temporal clustering structure, according to their low-dimensional latent trajectories. Although the proposed bi-clustering method is to resolve problems in neuroscience, this method can be potentially useful to extract insightful latent structures (bi-clustering and trajectories) from general large-scale (counting) time series.

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

# A APPENDIX

## A.1 MCMC UPDATES

The posteriors of the model parameters are sampled by a MCMC algorithm. To illustrate the sampling steps for each iteration, we expand them into six steps and provide the details as follows.

### A.1.1 UPDATE LATENT FACTORS

Denote the p.m.f. for $\text{NB}(r, \mu)$ as

$$f_{NB}(y \mid r, \mu) = \frac{\Gamma(r+y)}{y!\Gamma(r)} \left(\frac{r}{r+\mu}\right)^r \left(\frac{\mu}{r+\mu}\right)^y$$

, then the full conditional distribution of $\tilde{\boldsymbol{X}}_{1:T}$ is as follows:

$$P(\tilde{\boldsymbol{X}}_{1:T} \mid \{y_{i,1:T}, d_i.\boldsymbol{c}_i\}_{i=1}^N, \{\boldsymbol{b}_l, \boldsymbol{A}_l, \boldsymbol{Q}_l\}, \xi_t)$$

$$\propto \left(\prod_{i=1}^N \prod_{t=1}^T f_{NB}(y_{it} \mid d_i, \mu_{it})\right) \exp\left(-\frac{1}{2}\|\tilde{\boldsymbol{X}}_1\|_2^2\right) \prod_{t=2}^T \exp\left(-\frac{1}{2}\boldsymbol{s}_t'|\boldsymbol{Q}_{\xi_t}|^{-1}\boldsymbol{s}_t\right),$$

, where $\mu_{it} = \exp\left(d_i + \tilde{\boldsymbol{c}}_i'\tilde{\boldsymbol{x}}_t^{(z_i)}\right)$ and $\boldsymbol{s}_t = \tilde{\boldsymbol{X}}_t - \boldsymbol{A}_{\xi_t}\tilde{\boldsymbol{X}}_{t-1} - \boldsymbol{b}_{\xi_t}$. The full conditional distribution has no closed form, and we sample it via PG augmentation technique, i.e., by introducing the auxiliary PG variable, the transformed "effective" observation $\hat{y}_{it}$ performs like Gaussian, and hence the regular sampling algorithm like forward-filtering-backward-sampling (FFBS) can be implemented to sample $\tilde{\boldsymbol{X}}_{1:T}$. Specifically, we can sample $\tilde{\boldsymbol{X}}_{1:T}$ from full conditionals in two steps:

- Step 1: sample the auxiliary PG parameter $\omega_{it}$ and calculate the transformed response $\hat{y}_{it}$. For $i = 1, \ldots, N$ and $t = 1, \ldots T$, sample the $\omega_{it}$ from a Pólya-Gamma distribution as $\omega_{it} \sim \text{PG}\left(r_i + y_{it}, d_i + \tilde{\boldsymbol{c}}_i'\tilde{\boldsymbol{x}}_t^{(z_i)} - \log r_{it}\right)$. Then we can calculate the transformed response as $\hat{y}_{it} = \omega_{it}^{-1}\kappa_{it}$, where $\kappa_{it} = (y_{it} - r_i)/2 + \omega_{it}(\log r_i - d_i)$

- Step 2: sample the $\tilde{\boldsymbol{X}}_{1:T}$ by FFBS. Since the transformed response $\hat{y}_{it} \sim N(\tilde{\boldsymbol{c}}_i'\tilde{\boldsymbol{x}}_t, \omega_{it}^{-1})$, we can use the regular FFBS algorithm for Gaussian state-space model, the detailed algorithm can be found in Chapter 4 in Prado & West (2010).

For data doesn't have severe over-dispersion, it can be useful to assume Poisson distributed observations (i.e. assume $y_{it} \sim \text{Poisson}(\mu_{it})$), for ease of the model inference. The Poisson distribution doesn't fall within the PG augmentation scheme, and we cannot use the method described here. However, motivated by the fact that NB distribution approximates to Poisson distribution as $r$ goes to infinity, we can use the sample from NB model as the proposal, and add one more Metropolis-Hasting step to accept or reject it as in (Wei et al., 2023).

To ensure the model identifiability, we project the posterior samples to the constraint space, such that 1) $\tilde{\boldsymbol{x}}^{(j)}$ is zero-centered, i.e. $\sum_{t=1}^T \tilde{\boldsymbol{x}}_t^{(j)} = \boldsymbol{0}$ and 2) $\tilde{\boldsymbol{x}}_{1:T}^{(j)}\tilde{\boldsymbol{x}}_{1:T}'^{(j)}$ is diagonal, where $\boldsymbol{x}_{1:T}^{(j)} = (\tilde{\boldsymbol{x}}_1^{(j)}, \ldots, \tilde{\boldsymbol{x}}_T^{(j)}) \in \mathbb{R}^{(p+1)\times T}$ (Fokoué & Titterington, 2003). However, the model is still not identifiable in terms of the signed-permutation. For deterministic algorithm, we can put constraints based on singular value decomposition (SVD) as in Miller & Carter (2020), but this is not appropriate for the sampling algorithm. To resolve this, we simply search for the signed-permutation that has the closest Euclidean distance to reference trajectories (in this implementation, we use sample in the 100-th iteration. Before that, align current step to previous). Instead of fixing the reference trajectories manually, we may find the optimized value recursively using the Varimax Rotation-Sign-Permutation (Varimax-RSP) algorithm developed by (Papastamoulis & Ntzoufras, 2022).

### A.1.2 UPDATE SUBJECT BASELINE AND FACTOR LOADING

The sampling of subject baseline $d_i$ and factor loading $\boldsymbol{c}_i$ from full conditional distribution is regular NB regression problem for each neuron, which is again updated by PG augmentation technique. The idea is the same as in sampling latent factors, i.e., transform the spikes $y_{it}$ to be Gaussian like

by introducing the augmented parameters $\omega_{it} \sim \text{PG}\left(r_i + y_{it}, \mu_t^{(z_i)} + (1, \boldsymbol{x}_t^{'(z_i)})(d_i, \boldsymbol{c}_i')' - \log r_i\right)$. Therefore, $\hat{y}_{it} = \omega_{it}^{-1}\kappa_{it} \sim \mathcal{N}((1, \boldsymbol{x}_t^{'(z_i)})(d_i, \boldsymbol{c}_i')', \omega_{it}^{-1})$, where $\kappa_{it} = (y_{it} - r_i)/2 + \omega_{it}(\log r_i - \mu_t^{(z_i)})$.

### A.1.3 UPDATE DISPERSION

The dispersion for each neuron $r_i$ is updated via a Gibbs sampler, since the gamma distribution is the conjugate prior to it, under the compound Poisson representation. Specifically, let $p_{it} = \mu_{it}/(\mu_{it} + r_i)$, then the conditional posterior of $r_i$ is $\text{Gamma}\left(a_0 + \sum_{t=1}^T L_t, 1/(h - \sum_{t=1}^T \log(1 - p_{it}))\right)$, where $L_t \sim \text{Poisson}(-r_i \log(1 - p_{it}))$. Refer to Zhou et al. (2012) for more technical details such as how to sample $L_t$.

### A.1.4 UPDATE DISCRETE STATES

To update the discrete states $\xi_t$, we use the weak-limit Gibbs sampler for sticky HDP-HMM as in (Fox et al., 2008a), by constructing a finite approximation to the HDP transitions with finite Dirichlet distribution. This is motivated by the fact the infinite limit of hierarchical mixture model converges in distribution to a true HDP as $M \to \infty$, such that

$$\beta \sim \text{Dir}(\eta/M, \ldots, \eta/M),$$
$$\psi_l \sim \text{Dir}(\alpha\beta_1, \ldots, \alpha\beta_L).$$

By using this weak limit approximation, we can update the states by an efficient, blocked sampling algorithms. Refer to Fox et al. (2008b) for more details.

### A.1.5 UPDATE LINEAR DYNAMICS

Because of the Gaussian assumption in the model, updating the linear dynamics for each state is a Bayesian multivariate linear regression problem. Specifically, let $\tilde{\boldsymbol{X}}_{1:T}^* = (\mathbf{1}_T, \tilde{\boldsymbol{X}}_{1:T})$, then for state $l$ assume the conjugate priors for $\{\boldsymbol{b}_l, \boldsymbol{A}_l, \boldsymbol{Q}_l\}$ as

$$\boldsymbol{Q}_l \sim \mathcal{W}^{-1}(\boldsymbol{\Psi}_0, \gamma_0),$$
$$\text{vec}((\boldsymbol{b}_l, \boldsymbol{A}_l')) \sim \mathcal{N}(\text{vec}(\boldsymbol{B})_0, \boldsymbol{Q}_l \otimes \Gamma_0^{-1})$$

, where $\mathcal{W}^{-1}$ denotes the inverse-Wishart distribution. The priors are set as $\boldsymbol{\Psi}_0 = 0.01\boldsymbol{I}_{p+1}$, $\gamma_0 = (p+1) + 2$ and $\boldsymbol{B}_0 = (\mathbf{0}, \boldsymbol{I})'$. If there are $Q-$chunks of latent factors having $\xi_t = l$, and the denote the time steps for $q-$th chunk as $\tau_q - k_q : \tau_q$. Then, the full conditional distribution are:

$$\boldsymbol{Q}_l \mid \tilde{\boldsymbol{X}}_{1:T} \sim \mathcal{W}^{-1}(\Psi_n, \gamma_n),$$
$$\text{vec}((\boldsymbol{b}_l, \boldsymbol{A}_l')) \mid \tilde{\boldsymbol{X}}_{1:T} \sim \mathcal{N}(\boldsymbol{B}_n, \boldsymbol{Q}_l \otimes \Gamma_n^{-1})$$

, where

$$\Psi_n = \Psi_0 + \sum_{q=1}^Q S_q' S_q + (\boldsymbol{B}_n - \boldsymbol{B}_0)'\Gamma_0(\boldsymbol{B}_n - \boldsymbol{B}_0),$$

$$S_q = \tilde{\boldsymbol{X}}_{\tau_q - k_q + 1:\tau_q} - \tilde{\boldsymbol{X}}_{\tau_q - k_q:\tau_q - 1}^* \boldsymbol{B}_n$$

$$\gamma_n = \gamma_0 + \sum_{q=1}^Q k_q$$

$$\boldsymbol{B}_n = \Gamma_n^{-1}(\sum_{q=1}^Q (\tilde{\boldsymbol{X}}_{\tau_q - k_q:\tau_q - 1}^{'*} - \tilde{\boldsymbol{X}}_{\tau_q - k_q + 1:\tau_q}) + \Gamma_0 \boldsymbol{B}_0)$$

$$\Gamma_n = \sum_{q=1}^Q (\tilde{\boldsymbol{X}}_{\tau_q - k_q:\tau_q - 1}^{'*} \tilde{\boldsymbol{X}}_{\tau_q - k_q:\tau_q - 1}^*) + \Gamma_0$$

### A.1.6 UPDATE SUBJECT CLUSTER ASSIGNMENTS

To update the subject cluster labels $z_i$s, we use a partition based algorithm, similarly to Miller & Harrison (2018). Let $\mathcal{C}$ denote a partition of neurons, and $\mathcal{C}\backslash i$ denote the partition obtained by removing neuron $i$ from $\mathcal{C}$.

1. Initialize $\mathcal{C}$ and $\tilde{\boldsymbol{X}}_{1:T}^{(c)} : c \in \mathcal{C}\}$ (e.g., one cluster for all neurons in our simulation).
2. In each iteration, for $i = 1, \ldots, N$: remove neuron $i$ from $\mathcal{C}$ and place it:

   (a) in $c \in \mathcal{C}\backslash i$ with probability $\propto (|c| + \gamma)M_c(y_{i,1:T})$, where $\gamma$ is the hyperparameter of the Dirichlet distribution in equation 3 and $)M_c(y_{i,1:T})$ denotes the marginalized likelihood of neuron $i$ in cluster $c$, when integrating the loading $\tilde{c}_i$ out.

   (b) in a new cluster $c^*$ with probability $\propto \gamma \frac{V_n(s+1)}{V_n(s)} M_{\boldsymbol{\theta}^{(c^*)}}(\boldsymbol{y}_i)$, where $s$ is the number of partitions by removing the neuron $i$ and $V_n(s) = \sum_{l=1}^{\infty} \frac{l_{(s)}}{(\gamma l)^{(n)}} f_k(l)$, with $x^{(m)} = x(x+1)\cdots(x+m-1)$, $x_{(m)} = x(x-1)\cdots(x-m+1)$, $x^{(0)} = 1$ and $x_{(0)} = 1$.

The update is an adaptation of partition-based algorithm for DPM (Neal, 2000), but with two substitutions: 1) replace $|c_i|$ by $|c_i| + \gamma$ and 2) replace $\alpha$ by $\gamma V_n(t+1)/V_n(t)$. See more details and discussions in (Miller & Harrison, 2018).

Instead of evaluating the full likelihood, we integrate the subject-specific factor loading $\tilde{c}_i$ out to obtain the marginalized likelihood $M_c(y_{i,1:T})$ to achieve better mixing for high dimensional situation. The marginalized likelihood is evaluated by the Laplace approximation as follows:

$$M_c(y_{i,1:T}) \overset{\propto}{\sim} \prod_{t=1}^{T}(f_{NB}(y_{it} \mid r_i, \hat{\mu}_{it}^{(c)}))\pi(\hat{c}_i)|\hat{\Sigma}_{c_i}|^{1/2}$$

$$\hat{\mu}_{it}^{(c)} = d_i + \mu_{it}^{(c)} + \hat{c}_i'\boldsymbol{x}_t^{(c)}$$

, where $\overset{\propto}{\sim}$ means "approximately proportional to", $\hat{c}_i$ and $\hat{\Sigma}_{c_i}$ are MLE estimates and corresponding variance estimates (inverse of the negative Hessian at $\hat{c}_i$) from NB regression on $\boldsymbol{c}_i$.

Since we need to align the latent trajectories to reference value for each cluster, it's necessary to handle the label switching problem for cluster assignment. Here, we implement the Equivalence Casses Representative (ECR) alforithm (Papastamoulis & Iliopoulos, 2010), by setting the cluster assignment at the 100-th iteration (before 100-th step, align current allocation to previous step) as the pivot allocation. To relax the dependency on setting pivot manually, we can further consider iterative versions by Rodríguez & Walker (2014).

### A.2 SUPPLEMENTARY RESULTS FOR SIMULATIONS

Here, we show trace plots (Figure 4) of several parameters for NB bi-clustering model in simulation (section 3 and Figure 2). Visual inspection and Geweke diagnostics don't show significant issues with convergence.

### A.3 SUPPLEMENTARY RESULTS FOR NEURAL PIXELS

In this section, we first provide histograms and trace plots for some parameters in the MCMC chain for NB bi-clustering model (chain 1 of NB bi-clustering model) in shown in Figure 3. Again, visual inspection and Geweke diagnostics don't show significant issues with convergence.

We then provide results from two independent chains for both NB and Poisson bi-clustering model (i.e., four chains in total). Specifically, we show the similarity matrices of subject clusters sorted by maxPEAR estimates (Figure 6A) and by maxPEAR with anatomical sites (Figure 6B). These orders are obtained from first chain of NB model, which is the same order as used in Figure 3C. We further show the similarity matrices for state cluster for these four chains. Overall, these results show that NB model provides different clustering results compared to the Poisson one.

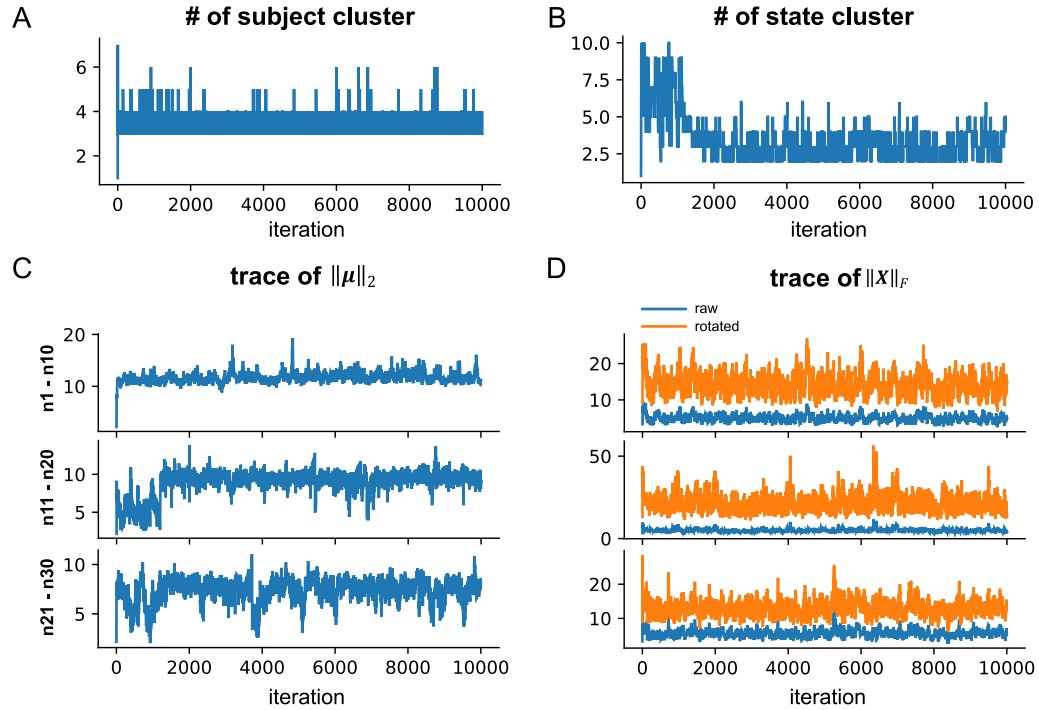

Figure 4: **Trace plots for simulations.** Here we show trace plots of several parameters for NB bi-clustering model in simulation (section 3 and Figure 2), including number of subject cluster (**A**) and number of state cluster (**B**). We further show the traces of $||\mu_{1:T}^{(j)}||_2$ (**C**) and $||\boldsymbol{x}_{1:T}^{(j)}||_F$ (**D**) for the 3 detected clusters (equivalent to true subject cluster). The way we define detected cluster is illustrated in section 3 and Figure 2E. For $||\boldsymbol{x}_{1:T}^{(j)}||_F$, the traces for both the raw (blue) and rotated (orange) samples are shown here

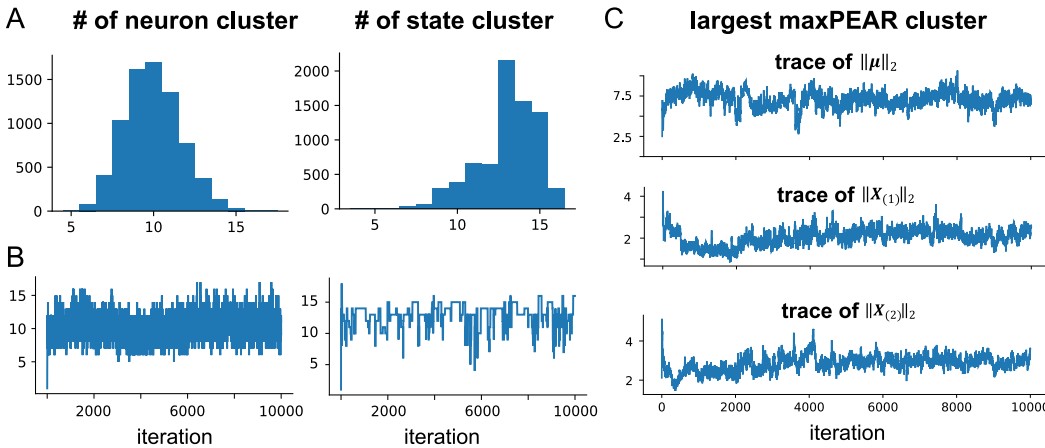

Figure 5: **Histograms and trace plots for chain 1 of NB-biclustering. A**. Histograms for number of neuron/ state cluster, using the samples from iteration 2500 to 10,000. The corresponding trace plots are shown in panel **B**. We then show the L2 norm for each latent trajectory, i.e. $\boldsymbol{\mu}$, $\boldsymbol{X}_{(1)}$ and $\boldsymbol{X}_{(2)}$, for the largest maxPEAR cluster, corresponding to latent trajectories in Figure 3E. Here $\boldsymbol{\mu}$ denotes $\mu_{1:T}$, $\boldsymbol{X}_{(i)}$ denotes $i$-th row of $\boldsymbol{x}_{1:T}$.

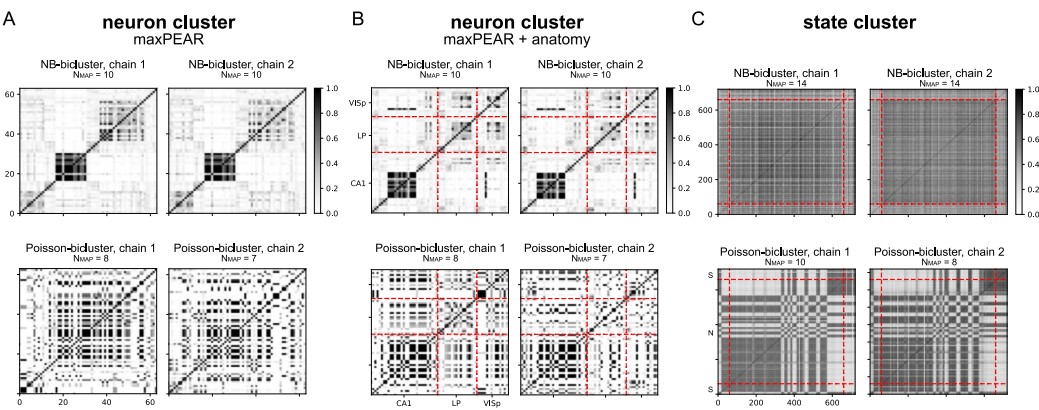

Figure 6: **Supplementary results for the application to Neuropixels data. A**. similarity matrices for neuron cluster, for 2 independent MCMC chains each on both NB and Poisson bi-clustering models. These matrices are sorted by maxPEAR estimates of chain 1 for NB model (also used in main text, Figure 3C i). **B**. We further sorted the matrices by anatomical sites, which is the same order used in the main text, Figure 3C ii-iv. C. The similarity matrices for state cluster, for 2 independent chains on each model.

