# OpenReview forum: "Bayesian Bi-clustering of Neural Spiking Activity with Latent Structures"
_ICLR.cc/2024/Conference — ICLR 2024 poster_

### Official Review · Reviewer_DWGp · 2023-10-29

**Soundness:** 2 fair
**Presentation:** 3 good
**Contribution:** 2 fair
**Rating:** 6
**Confidence:** 3

**Summary:**

This paper proposes a spatio-temporal clustering method to analyze multiple neural populations. This method could find the clusters in neurons (spatial) and different states over time (temporal). To flexibly extract the bi-clustering structure, the authors model the spike data in a non-parametrical way, where the subject clustering structure is modeled by a mixture of finite mixtures model and the state clustering structure is modeled by a sticky Hierarchical Dirichlet Process Hidden Markov Model. The inference is performed by MCMC with the Polya-Gamma technique. In the experiments, the authors evaluate their method on both simulated data and neural recordings.

**Strengths:**

* Different from previous LDS methods, this paper explores multi-region neural data from a new perspective: the authors try to understand multi-neural populations by spatiotemporal clustering structures.

* Non-parametrically model the neural data so that there is no need to prespecify the number for subject and state clusters.

**Weaknesses:**

* No analysis of the scalability of this method. For both syntactic data and real neural data, the number of neurons is small (e.g., 30, 60). Could this method generalize to a large neural recording? Such as a larger number of neurons and a longer time stamps.

* No comparison of the proposed model with other latent variable models like SLDS and rSLDS.

* Some typos, e.g., the "cite" doesn't refer to a paper in section 2.3, the figure index should be 2 rather than 3 in section 3, and the figure index should be 3 rather than 4 in section 4.

**Questions:**

*  What's the time complexity (concerning the number of iterations, trials, neurons, and time points) of the proposed model with the efficient MCMC algorithm?

* What kind of desirable neural data could be better analyzed by such spatiotemporal clustering structures rather than LDS-based models (e.g., SLDS, r-SLDS)?

* Could you please discuss the relationship of this method to the PP-Seq model [1]? PP-Seq is an unsupervised non-parametrical model to detect neural sequences in high dimensional neural recordings, which could be considered as a clustering approach to finding spatiotemporal neural patterns.

[1] Williams, Alex, et al. "Point process models for sequence detection in high-dimensional neural spike trains." Advances in neural information processing systems 33 (2020): 14350-14361.

---

> ### Author Response · Authors · 2023-11-12
>
> Thanks a lot to the reviewer for their positive comments. Here, we clarified some specific points…
>
> 1. > What's the time complexity...
>
>     Thanks for suggestions on studying time complexity more carefully. The simulation is fast, but for this application example, it takes around 1s for each iteration with brute force implement in a i7-8665U CPU @ 1.90GHz 2.11 GHz laptop. To make it scalable for large dataset, besides implementation of computation techniques like parallel, we can instead do variational inference (VI). But before infer model by VI, checking by MCMC is always good :).
>
> 2. > Some typos...
>
>     Thanks, just fixed.
>
> 3. >... rather than LDS-based models (e.g., SLDS, r-SLDS)?
>
>     The proposed model is also a LDS-based model, and here we use the latent trajectories to define neural and temporal clusters, so that the detected clusters can be interpretaed as "functional".
>
>     When implementing the mentioned model SLDS, r-SLDS to multiple population, we need to first define appropriate "neural population". This is hard but important (e.g. neither anatomy nor cell type may be appropriate, since neurons may 'cross talk'), and bad definition on "neural population" can mess up estimation of latent trajectoreis. For example, in C.elegens application in population-rSLDS [1], they deal "with different notions of population structure" and they claim that "there are many reasonable definitions of population ". So, it would be interesting to avoid these confusions, and define "functional" group by latent states. (Also, doing everything together can avoid selective inference issue)
>
> 3. > Could you please discuss the relationship of this method to the PP-Seq model? ...
>
>     Detecting neural sequence is an interesting problem and PP-Seq works perfectly for this. However, the problems we want to solve are not exactly the same. Basically...
>
>     - PP-Seq tries to detect neural sequence, and the "clustering element" is spikes at each point. Therefore, for each neuron, it can belong to different clusters at different time (red/ blue sequences in PP-Seq paper can be interleaved), and also there will be different group across neuron at the same time.
>     - However, here we want to cluster neurons over time and find the synchronous state changes.  We factorize spatial and temporal cluster structure. For subject cluster, the "clustering element" is the whole trajectory, and each neurons consistently belong to only 1 neural cluster. While for temporal cluster, the state is synchronous across all neurons. Figure 1A can give you a better sense of the difference.
>     - The neural sequence defined by PP-Seq based on several "attributes", and attribute like amplitude may not be appropriate in some cases (e.g. firing of pre-neuron may inhibit post-neuron). Here all clustering structure are based on latent trajectories, i.e., the pattern of the waveform, which can have "functional" interpretation.
>
>     Also thanks for mentioning PP-Seq, and it can be valuable to take seqeuntial response (delay of response) into account when clustering, by modeling the latent factors in a more careful way.
>
> [1] Joshua I. Glaser, Matthew Whiteway, John P. Cunningham, Liam Paninski, and Scott W. Linderman. 2020. Recurrent switching dynamical systems models for multiple interacting neural populations. In Proceedings of the 34th International Conference on Neural Information Processing Systems (NIPS'20). Volume 33, 14867–14878.

---

### Official Review · Reviewer_xUve · 2023-11-01

**Soundness:** 3 good
**Presentation:** 3 good
**Contribution:** 3 good
**Rating:** 8
**Confidence:** 5

**Summary:**

Motivated by modeling of neural spiking activities, the paper proposes a bi-clustering approach at both spatial and temporal levels. Individual count data are modeled through negative binomial regression. The spatial (subject) clusters are modeled through mixture of finite mixtures. The temporal (state) clusters are modeled through hierarchical Dirichlet process. The dependency of negative binomial regression on the temporal dimension is given by a hidden Markov model. A MCMC algorithm is designed for sampling from the posterior distribution. Both synthetic numerical experiments and real applications are provided.

**Strengths:**

1. The problem is well-motivated with a meaningful and important application.
2. The paper is mostly well-written and clearly presented.
3. Sufficient background and preliminaries are provided.
4. Details on the derivation of the MCMC algorithms are provided.
5. The challenging goal of conducting full Bayesian inference for a complex clustering task is of itself great importance.

**Weaknesses:**

1. While some constraints required for identifiability are provided at the end of section 2.1, I am not convinced that these are sufficient conditions. A theoretical proof of the model identifiability along with all necessary conditions seems important here, given the vast number of parameters.
2. The MCMC algorithm has incorporated all the most modern efficient MCMC techniques, including the Polya-Gamma augmentation, Miller&Harrison sampler for mixture of finite mixtures, FFBS for state space models, etc. The effort here is worth being recognized. However, for both the simulation and application, the MCMC sampler is run for only 1000 iterations. I am concerned of whether the MCMC sampler has really converged given its complexity and the vast amount of model parameters. If it has indeed converged, it would be nice to present trace plots of parameters and also conduct MCMC convergence diagnosis (e.g. using Gelman-Rubin or Geweke statistics).
3. There are some typos in the paper and some terms undefined, e.g.
    - the distribution \mathcal{S} in equation (4) is only given in the appendix
    - in the first paragraph of Sec 2.1, where \log\mu_{i, t} = d_i + \tilde{c}' \tilde{x}_t^{(z_i)} has missing subscript i in \tilde{c}.

The general idea of this paper and the designed MCMC sampler are both quite nice. I would be willing to raise the rating by one or two levels if the points 1 and/or 2 can be fully addressed.

**Questions:**

The temporal clusters are captured in the temporal dynamic in the AR(1) model. What is the motivation of such modeling from an application perspective, compared to simpler modeling (e.g. directly model temporal clusters of \tilde{X}_t through change points, etc)?

---

> ### Author Response · Authors · 2023-11-12
>
> Thanks a lot to the reviewer for their positive comments. Here, we clarified some specific points…
>
> 1. > While some constraints required for identifiability...
>
>     The model idetifiability issue has been studied for a long time in factor analysis (FA) model (if we ignore the temporal system equation and do matrix transpose, the model is equivalent to Poisson FA), and we discuss details in appendix A.1.1. Originally, people claim that constraining $1/2p(p-1)$ parameters and using diagonal constrains is enough (reivew in [1]), there are still permutation and sign issue. To further resolve this, we simply search for the signed-permutation that has the closest Euclidean distance to the previous sample. A more "MCMC"-type solution is developed in [2]. Since in our model, we factor the bias $d_i$ out, we need further "centering" constraint. Besides these conceptual and theoretical claims, empirically we don't find problems in our MCMC.
>
> 2. > ...However, for both the simulation and application, the MCMC sampler is run for only 1000 iterations...
>
>     Thanks for the compliment. Here, we add the trace plots for both simulation and application in the appendix. Visually, the trace plots don't show significant problems. Intuitively, the fast convergence may mainly result from the model assumption, where AR(1) structure in system equation hugely decrease the "effective" dimension of parameters. However, for formal analysis, 1000 iterations is far less than enough and we definitely need run longer chains.
>
>     The trace plot for state number also matches to our statement in the 3rd point for improvement in discussion section. The mixing for state cluster is not very good (we tried with running the HDP-HMM [3] and HDP-HSMM [4] when fixing other parameters, and the mixing is still not perfect), and imporving sampling algorithm for HDP-HMM/HSMM can be an interesting direction (some potential ideas in discussion).
>
> 3. > There are some typos in the paper...
>
>     Thanks, just fixed.
>
> 4. > What is the motivation of such modeling from an application perspective....
>
>     Good question! For many problems (although not for this paper), we may be more interested in the transition matrix $A_{\xi_t}$ (maybe also $Q_{\xi_t}$), since it summarizes the interactions between different neural populations. Therefore, clustering based on dynamics parameters $(b_{\xi_t}, A_{\xi_t}, Q_{\xi_t})$ can provide us the dynamics for interactions between neural population. Simpler modeling is good enough for temporal clustering, and I guess the results can be very similar. But we may lose some insights on dynamics for neural population interactions.
>
> [1] Fokoué, E., Titterington, D. Mixtures of Factor Analysers. Bayesian Estimation and Inference by Stochastic Simulation. Machine Learning 50, 73–94 (2003). https://doi.org/10.1023/A:1020297828025
>
> [2] Papastamoulis, P., Ntzoufras, I. On the identifiability of Bayesian factor analytic models. Stat Comput 32, 23 (2022). https://doi.org/10.1007/s11222-022-10084-4
>
> [3] Emily B. Fox, Erik B. Sudderth, Michael I. Jordan, and Alan S. Willsky. 2008. An HDP-HMM for systems with state persistence. In Proceedings of the 25th international conference on Machine learning (ICML '08). Association for Computing Machinery, New York, NY, USA, 312–319. https://doi.org/10.1145/1390156.1390196
>
> [4] Matthew J. Johnson and Alan S. Willsky. 2013. Bayesian nonparametric hidden semi-Markov models. J. Mach. Learn. Res. 14, 1 (January 2013), 673–701.

---

> > ### Comment · Reviewer_xUve · 2023-11-14
> >
> > Thanks a lot for the detailed answers.
> >
> > For the model identifiability, constraining the parameters and doing postprocessing to handle the label switching problem seems pretty reasonable.
> >
> > The MCMC trace plots seems pretty standard to what we generally see in large models with both continuous and discrete parameters. The mixing is fine, but it appears that at least 10,000+ MCMC samples should be used in your final version.
> >
> > I have increased my ratings accordingly.

---

> > > ### Author Response · Authors · 2023-11-14
> > >
> > > Thank you so much & really an enjoyable discussion!

---

### Official Review · Reviewer_96hg · 2023-11-06

**Soundness:** 3 good
**Presentation:** 3 good
**Contribution:** 3 good
**Rating:** 6
**Confidence:** 4

**Summary:**

This paper proposes a nonparametric Bayesian biclustering algorithm to simultaneously form spatial (neuron) clusters and temporal clusters for multiple (count) time series. The algorithm is carried out through efficient MCMC sampling and is shown to be able to recover all model parameters in simulation studies. The proposed algorithm is then applied to a real data set to show its effectiveness.

**Strengths:**

The algorithm is clearly described, reasonable assumptions are imposed, and it has a potentially large range of applications.
The overall writing is good and the presentation is clear.

**Weaknesses:**

The numerical results, including simulation studies and real data application, are not enough convincing.

**Questions:**

1. The authors only reviewed some work in time series clustering. However, the neural spiking activity data is originally a point process and one has to aggregate them using small bins into count time series. There is some existing work on finding clusters in point process literature as well, which in my opinion should be reviewed for relevance. For example, [1], [2], [3], and references therein.

2. The simulation study is a little bit too simple, with only one parameter setting, and there are no comparisons to the state-of-the-art methods.   I would suggest investigating three things: (1) the sensitivity of the bin sizes; (2) how the estimation performances change when the number of nodes and the Time length increases; (3) compared to some existing methods in terms of the predictive performances (one may consider using cross-validations.)

3. In the real data analysis, the authors argue that it is necessary to use the biclustering algorithm. However, no comparison to any existing method is given.  Similar to the simulation studies, I suggest comparing the predictive performance of the proposed algorithm to some existing methods.

4. In the simulation study, there is no histogram in Figure 3(c). Could you please clarify?

[1] Xu, H., & Zha, H. (2017). A dirichlet mixture model of hawkes processes for event sequence clustering. Advances in neural information processing systems, 30.

[2] Yin, L., Xu, G., Sang, H., & Guan, Y. (2021). Row-clustering of a Point Process-valued Matrix. Advances in Neural Information Processing Systems, 34, 20028-20039.

[3] Fang, G., Xu, G., Xu, H., Zhu, X., & Guan, Y. (2023). Group network Hawkes process. Journal of the American Statistical Association, (just-accepted), 1-78.

---

> ### Author Response · Authors · 2023-11-12
>
> Thanks a lot to the reviewer for their positive comments. Here, we clarified some specific points…
>
> 1. > The authors only reviewed some work in time series clustering. However,...
>
>     Thanks for menioning these methods and providing the refereces. All these methods are very great, but the defined "clusters" may not directly related to underlying mechanisms and may lose some scientific insights. This motivates us to define neuron and temporal states by latent trajectories directly, as latent factors can usually have scientific interpretation (more details in intro & discussion). We reviewed these methods and added several sentences on this at the begining of paragraph 3 in Intro section.
>
> 2. > The simulation study is a little bit too simple...Similar to the simulation studies, I suggest comparing the predictive performance of the proposed algorithm to some existing methods.
>
>     Thanks the reviewer for these constructive suggestions. All these simulation experiements can make our methods more convincing, especially for suggestions on sensitivity of bin size, since selection on bin size is important to balance computation time and resolution in practice. These are very interesting directions for future work, but the main goal here is to illustrate new idea, and extensive study may beyond the scope (and page limit) of this paper.
>
> 3. > In the simulation study, there is no histogram in Figure 3(c).
>
>     The histograms of cluster(s) number are dropped because of page limit and they are not main interest for application (more interested in clustering pattern). Now, we add the histogram & trace plot in the appendix for reference. Inference on neuron cluster number is "noisy", and it is common for real data analysis (similar to Figure 1 in MFMM paper [1]).
>
>  [1] Jeffrey W. Miller & Matthew T. Harrison (2018) Mixture Models With a Prior on the Number of Components, Journal of the American Statistical Association, 113:521, 340-356, DOI: 10.1080/01621459.2016.1255636

---

### Meta-Review · Area_Chair_spg8 · 2023-12-09

**Metareview:**

This paper presents a novel nonparametric Bayesian algorithm for biclustering multiple count time series. The algorithm simultaneously identifies groups of neurons (spatial clusters) and their corresponding states over time (temporal clusters). It achieves this by effectively sampling from the posterior distribution using efficient MCMC techniques, demonstrating its ability to accurately recover all model parameters in simulated studies. The effectiveness of the proposed algorithm is further validated through its application to a real-world dataset.

There are some notable weaknesses: limited numerical results, lack of convergence diagnostics, and missing analysis of scalability, comparison with existing methods, and proper error checking. While reviewers generally view the paper favorably, I have significant concerns about the experiment and result interpretation. Comparisons with other multi-region or state space models are missing, making it difficult to attribute the observed spatial and temporal clusters to the data or model artifacts. The analysis seems to assume the obtained results are correct without metrics to demonstrate the method's superiority or the unsupervised method's effectiveness in explaining the data. Figure 3D, specifically the maxPEAR states, raises concerns with its excessive state switching and the provided explanation attributing it to potential subgroups and state changes in the same experiment settings. These explanations appear obvious and lack the novelty and rigor expected of scientific findings.

Given these major concerns, I think the paper needs a lot of improvement before publication at ICLR.

Note by PCs: This paper was borderline and, based on SAC's recommendation, the decision is to accept the paper. We encourage the authors to take AC's feedback into account and improve the paper in the camera-ready version.

**Justification For Why Not Higher Score:**

N/A

**Justification For Why Not Lower Score:**

N/A

---

### Decision · Program_Chairs · 2024-01-16

Accept (poster)